Methods

# New algorithms for accurate and efficient de novo genome assembly from long DNA sequencing reads

Laura Gonzalez-Garcia[1], David Guevara-Barrientos[1], Daniela Lozano-Arce[1], Juanita Gil[2], Jorge Díaz-Riaño[1], Erick Duarte[1], Germán Andrade[1], Juan Camilo Bojacá[1], Maria Camila Hoyos-Sanchez[1], Christian Chavarro[1], Natalia Guayazan[3], Luis Alberto Chica[4,5], Maria Camila Buitrago Acosta[1], Edwin Bautista[1], Miller Trujillo[1], Jorge Duitama[1]

Building de novo genome assemblies for complex genomes is possible thanks to long-read DNA sequencing technologies. However, maximizing the quality of assemblies based on long reads is a challenging task that requires the development of specialized data analysis techniques. We present new algorithms for assembling long DNA sequencing reads from haploid and diploid organisms. The assembly algorithm builds an undirected graph with two vertices for each read based on minimizers selected by a hash function derived from the k-mer distribution. Statistics collected during the graph construction are used as features to build layout paths by selecting edges, ranked by a likelihood function. For diploid samples, we integrated a reimplementation of the ReFHap algorithm to perform molecular phasing. We ran the implemented algorithms on PacBio HiFi and Nanopore sequencing data taken from haploid and diploid samples of different species. Our algorithms showed competitive accuracy and computational efficiency, compared with other currently used software. We expect that this new development will be useful for researchers building genome assemblies for different species.

## Introduction

Contiguous and accurate assembly of complex eukaryotic genomes is one of the most challenging tasks in current biotechnology and bioinformatics (Baker, 2012; Nurk et al, 2022). Bioinformatic tools for genome assembly are used to sort and orient partial reads produced by various sequencing technologies. Partial genome assemblies, including most gene-rich regions, have been generated in the last decade. However, contiguous and high-quality assemblies are required to integrate synteny information in genome-scale comparative genomics and pangenomics, to study evolution and dynamics of mobile elements, for population genomic analysis, such as genome-wide association studies, and for the discovery of genomic footprints of selection (Amiri et al, 2018; Xu et al, 2020). High-quality assemblies are also useful to understand the genome evolution of species (Hu et al, 2021), to identify structural variations (Ouzhuluobu et al, 2020), and to define the gene repertoire including targets for resistance in plants and animals, and virulence factors and effectors in pathogens (Bhadauria et al, 2019). This complete gene catalog is key for identifying interesting genomic target regions for plant and animal breeding (Low et al, 2020; Song et al, 2021), and for personalized medicine. Large assembly efforts, such as that performed by the Vertebrate Genomes Project, highlight the importance of building high-quality genome assemblies (Rhie et al, 2021) to preserve genetic information of life. Moreover, genome assemblies have been useful in pathogen surveillance for public health (Taylor et al, 2019).

The production of sequencing data has grown exponentially in the last years, and genome assembly has become a routine task; however, most of the currently available genomes have been sequenced using high-quality short-read technologies such as Illumina. Currently, long-read technologies, such as PacBio and Nanopore, have improved the quality of data and allowed a better de novo assembly of genomes, haplotype phasing, and structural variant identification (Hon et al, 2020). Nanopore sequencing technologies offer the advantage of producing the longest read lengths (Mbp range), the more common lengths being 10–30 Kb, as these are limited by the quality and size of the DNA delivered to the sequencing pore (Amarasinghe et al, 2020). Furthermore, some of the Nanopore sequencers can be portable and generate data in real time, proving useful for field research and diagnostics (Xu & Seki, 2019). In contrast, following the continuous long read (CLR) protocol, PacBio single-molecule real-time sequencing delivers reads of 30 Kb on average, but it has a low coverage bias across different values of G + C content, and allows for the direct detection

[1]Systems and Computing Engineering Department, Universidad de los Andes, Bogotá, Colombia   [2]Department of Entomology and Plant Pathology, University of Arkansas, Fayetteville, AR, USA   [3]Department of Biological Sciences, Universidad de los Andes, Bogotá, Colombia   [4]Research Group on Computational Biology and Microbial Ecology, Department of Biological Sciences, Universidad de los Andes, Bogotá, Colombia   [5]Max Planck Tandem Group in Computational Biology, Universidad de los Andes, Bogotá, Colombia

Correspondence: ja.duitama@uniandes.edu.co

of DNA base modifications (Nakano et al, 2017). One important difficulty in the analysis of both Nanopore and PacBio CLR data was that the reads produced by both technologies initially had an average error rate of ~15%. Both platforms have made further efforts to reduce this error rate. One important step forward in the case of PacBio was the development of a new protocol, called circular consensus sequencing, to generate high-fidelity reads with an average error rate of ~0.5% (Wenger et al, 2019). This enabled the assembly of complete chromosomes, even for diploid or polyploid organisms. Despite the high assembly contiguity achieved with long reads, other strategies can be used to improve assemblies, such as Hi-C (Zhou et al, 2019; Kronenberg et al, 2021; Cheng et al, 2022), parental information (Koren et al, 2018), and Strand-seq (Hills et al, 2021). An important achievement in genome assembly, which combines these technologies, is the recent telomere-to-telomere assembly of the diploid HG002 human individual (Jarvis et al, 2022).

Most of the commonly used tools to assemble long-read datasets implement the overlap–layout–consensus (OLC) algorithm. These were developed to assemble reads with high error rates, such as the Nanopore and PacBio CLR reads. Canu (Koren et al, 2017) uses a MinHash overlapping strategy (Berlin et al, 2015) with a tf-idf weighting to identify overlaps. Then, a linear graph is constructed using a greedy best-overlap algorithm. WTDBG (Ruan & Li, 2019) implements minimizers for efficient identification of overlaps. Flye (Kolmogorov et al, 2019) implements an algorithm to resolve repeats from a possibly inaccurate initial assembly. FALCON (Chin et al, 2016) implements a simple haplotype phasing algorithm to perform read clustering and to generate phased assemblies. After the emergence of PacBio HiFi reads, new algorithms have been developed to perform error correction. These algorithms aim for perfect reads in which single-nucleotide differences can be used to resolve differences between repetitive elements (Nurk et al, 2020; Cheng et al, 2021). HiCanu is an improvement of Canu that implements homopolymer compression to align and correct reads having base counts on homopolymer tracts as main source of error (Nurk et al, 2020). Hifiasm integrates haplotype phasing to perform haplotype-aware error correction (Cheng et al, 2021). Error correction of long reads, especially Nanopore reads, remains an important step during genome assembly and is usually a computationally expensive process. NECAT was developed as an error corrector and de novo assembler for Nanopore reads (Chen et al, 2021). In NECAT, error correction is based on a two-step progressive method by which low-error-rate subsequences of reads are corrected first, and then, they are used to correct high-error-rate subsequences.

In this work, we introduce a new software implementation for genome assembly from long-read sequencing data. It includes new algorithmic approaches to build OLC assembly graphs and to identify layout paths. Benchmark experiments on PacBio HiFi and Nanopore data from organisms of different species including *Escherichia coli*, yeast, *Drosophila melanogaster*, rice, maize, and humans show that our algorithms are competitive and, in some cases, more accurate, compared with previous solutions. These algorithms are implemented as part of the Next Generation Sequencing Experience Platform (NGSEP) (Tello et al, 2019), allowing tight integration with genome comparison and detection of genomic variants within a single easy-to-use tool for analysis of both short- and long-read DNA sequencing data.

# Results

## k-Mer count–based hashing for efficient and accurate construction of assembly graphs

We implemented a new hashing scheme for minimizers to efficiently identify overlaps and build OLC graphs. Fig 1 shows the implemented algorithm to build an overlap graph and a layout. The graph construction is similar to that of the best-overlap graph (Miller et al, 2008), having two vertices for each read representing the start (5′-end) and the end (3′-end) of the read. In this representation, the graph does not need to be a multigraph. Let $X^s$ and $X^e$ be the two vertices generated from each read X. If the end of read A has an overlap with the start of read B, this overlap is represented with the edge $\{A^e, B^s\}$. Conversely, if the end of read A has an overlap with the start of the reverse complement of B, this overlap will be represented by the edge $\{A^e, B^e\}$. In our representation, the graph is completely undirected to take into account that reads are sequenced from the two strands of the initial template with equal probability, and hence, there is no a priori information on which one should be considered the positive strand.

Similar to the graph construction implemented in WTDBG (Ruan & Li, 2019), we built a minimizer table from the reads, to identify overlaps in linear time relative to the total number of sequenced base pairs. However, we implemented a different procedure to calculate hash codes that changes the priority to select k-mers as minimizers. Before calculating minimizers, we first build a 15-mer spectrum table, calculating the count distribution across the reads. Analyzing this distribution, the algorithm infers the mode that corresponds to the average read depth and estimates the assembly size. To achieve an efficient calculation of the k-mer distribution, the spectrum table is built with a fixed k-mer length of 15 (instead of the input k-mer length used later), because that is the maximum length to create the table as a fixed array of length $2^{30}$ in which the index of the array corresponds to a unique encoding of each possible DNA k-mer. The data type of this array is a two-byte integer to store a count per k-mer up to $2^{15}$, which is enough for real whole-genome sequencing datasets. This implementation ensures a fixed memory usage of $2^{31}$ bytes (about 2 gigabytes), regardless of the input size and genome complexity. The 15-mer spectrum allows not only to approximate the assembly length and average read depth, but also to calculate the hash value of read k-mers.

To identify overlaps, k-mers of a user-defined length (up to 31) are calculated for each read. Each k-mer is uniquely encoded as a 62-bit number $b$, and the count $x$ of the 15-mer suffix on the 15-mer spectrum is calculated. A rank $r(x)$ is calculated from the count, as two times the distance from the mode corresponding to the haploid number. The hash value $h(b)$ is calculated as the number of k-mers with rankings smaller than $r(x)$ plus the module of the division between $b$ and the smallest prime number larger than $x$. This last term is a simple scheme to simulate randomness for k-mers within the same rank. This hashing scheme allows the prioritization of real

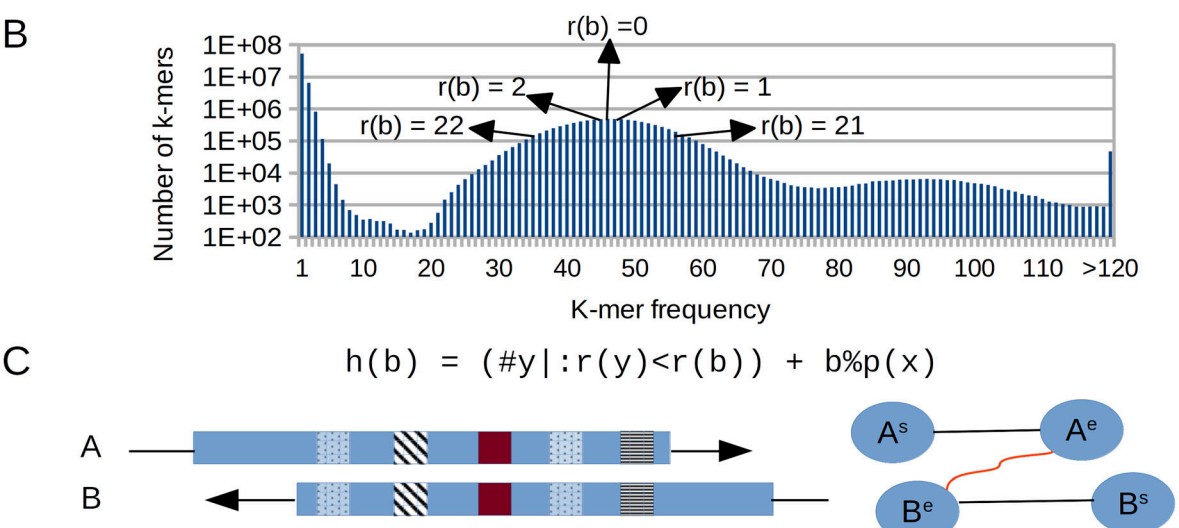

**Figure 1. Overview of the graph construction algorithm implemented in NGSEP for de novo assembly of long reads.**
**(A)** Fixed array to calculate counts of 15-mers. **(B)** Distribution of k-mer frequencies is used to rank edges based on their distance from the peak corresponding to single-copy regions. **(C)** Hash value is calculated from the rank to select minimizers and identify overlaps. Dynamic programming is used to cluster k-mer hits.

k-mers that are likely to come from single-copy regions of the haploid genome during the calculation of minimizers. At the same time, k-mers from repetitive regions have larger hash codes, which reduces their priority to become minimizers but does not discard them completely.

We implemented a simulated alignment of each candidate overlap to calculate different measures associated with each edge in the overlap graph, avoiding a complete pairwise alignment between candidate pairs at this stage of the process. First, matching k-mers (minimizers) between a subject (longer) read and a query (shorter) read are clustered based on consistency of the prediction of overlap start that can be inferred from the relative location of the k-mer in the subject sequence. Assuming that insertion and deletion errors have a similar probability of occurrence (mainly within homopolymer runs), the inferred starting point for k-mers corresponding to a real overlap should be consistent (have a low variance). Conversely, inferred starting points for matching k-mers supporting false-positive overlaps because of repetitive structures (up to a certain length) should have a larger variance. We implemented a clustering procedure similar to k-means to group k-mer hits that are likely to support the same alignment, using the inferred starting points as centroids. The average number of k-mer hits for each k-mer is used to infer the number of different clusters that can be expected. Up to two clusters with the largest k-mer count are retained as long as they support two of the four possible

alignment configurations (start–start, start–end, end–start, and end–end). Because an overlap length can also be inferred from each matching k-mer, the overlap for a cluster of matching k-mers is inferred as the average of the inferences performed from each matching k-mer.

**Layout construction as an edge selection problem**

The statistics collected during the simulated alignment step are used during the layout stage to select edges that will be part of the assembly paths. For each edge, derived from a k-mer cluster, relevant statistics include the predicted overlap, the number of shared k-mers building the overlap, the number of base pairs from the subject sequence covered by the shared k-mers (CSK), and the first and the last position of both the subject and the query sequences having k-mers supporting the possible overlap. The layout algorithm ranks and selects edges based on the knowledge that can be inferred from the distribution of the different statistics. Although in a real experiment true layout edges are unknown, we first identify edges that are reciprocal best for their corresponding vertices, in terms of both overlap length and CSK, and that connect vertices with a total degree less than three standard deviations from the average. These edges are termed "safe," and it is assumed that they will be part of the layout. Because they are reciprocal best, these edges will generate an initial series of paths within the graph.

Moreover, it is assumed that the distribution of overlap length and CSK calculated from these edges would be a good representation of the distributions calculated from all true layout edges. The cost of each remaining edge is calculated as a likelihood of the edge features given the distributions inferred from the safe edges. Whereas a normal distribution is fitted for the overlap and the CSK, a beta distribution is fitted for the proportion of overlap calculated from the first and the last overlap positions supported by k-mers. Likelihoods are calculated as *P*-values of the edge features. Log likelihoods of the features are added to calculate the total edge likelihood and sort edges based on this feature. Edges are then traversed in a descending order to augment the paths initially derived from safe edges. An edge is selected if it does not include an internal path vertex and if it does not create a cycle. Fig 2 shows a schematic diagram of this procedure.

Once paths are constructed, an initial consensus is built concatenating layout vertices. On each step, the next read is aligned to the consensus end to recalculate the true overlap and the consensus is augmented with the substring corresponding to the overhang of the alignment. At the same time, embedded reads are recovered and mapped to the consensus contigs. In order to improve the per base quality of the assembly, once all input reads are mapped to the assembly, the following polishing algorithm is executed on the aligned long reads: first, pileups are calculated for each position to identify the base with the largest count and update the consensus if needed. Then, similar to the process to call variants, a second step calculates "active regions" across the alignment, which are defined as contiguous regions in which each base pair is at most 5 bp away from an indel call. Once active regions are calculated, a de Bruijn graph is built from the read segments spanning the active region and a mini-assembly is executed to calculate the corrected segment.

### Benchmark with PacBio HiFi data

To test the performance of NGSEP with PacBio HiFi data, we assembled genomes from publicly available HiFi reads of the indica rice variety Minghui63 (15- and 20-kbp reads), the B73 maize inbred line, and the human haploid cell line CHM13 using NGSEP and three commonly used tools (Canu, Flye, and Hifiasm). Fig 3 shows the results of these benchmark experiments. The contiguity of each assembly, measured as the Nx curve, is contrasted with the number of misassemblies against a curated reference genome, as measured by Quast (Gurevich et al, 2013). The complete statistics are available in Table S1.

Regarding the rice data, the assemblies generated by Hifiasm and NGSEP have the highest N50 values for the 15- and 20-kbp datasets, respectively. In both cases, at least 95% of the genome (395 Mbp) was assembled in less than 20 contigs. Canu ranks third, close to NGSEP for the 15-kbp dataset and close to Hifiasm for the 20-kbp data. Flye shows the lowest contiguity in all datasets (Fig 3A). N50 and NG50 values were identical for all rice assemblies with the exception of the Canu assembly for the 15-kbp dataset and the Hifiasm assemblies (Fig S1). The reason behind the observed differences is that these assemblies have a total length between 20 Mbp and 50 Mbp greater than the length of the reference (Table S1).

Conversely, for the maize and the CHM13 datasets, the assemblies generated by NGSEP have a lower contiguity compared with those generated by Hifiasm and Canu; however, they have better contiguity compared with the assemblies generated using Flye (Fig 3A). For the maize dataset, all the tools assembled the genome in more than 500 contigs with a minimum length of 50 kbp. In these assemblies, the N50 value ranged from 6.1 Mbp (Flye) to 37.5 Mbp (Hifiasm). The lower contiguity of the NGSEP and Flye assemblies, compared with those of Canu and Hifiasm is probably caused by a lower average read length (12 kbp), lower read depth (23×), and higher genome complexity, compared with the rice datasets. The same ranking was observed in the human cell line, which was sequenced at a mean read length of 15 kbp and read depth of 33×. The higher read depth yielded an improvement of at least 20 Mbp in N50 values for all assemblers. In this case, the N50 value ranged from 29.1 Mbp (Flye) to 86.8 Mbp (Hifiasm). In both samples (maize and CHM13), the NG50 values of the Canu assemblies were longer than the N50, again because of the difference between assembly length and the reference genome length. Conversely, the NG50 is between 1 Mbp and 4 Mbp smaller for the CHM13 assemblies generated by NGSEP, Flye, and Hifiasm.

Fig 3B shows the number of misassembly errors identified by Quast, using a curated reference genome for comparison. Errors are classified as long-range misassemblies (m1) and local misassemblies (m2). The proportion of m1 errors, relative to the total, ranged from 0.15 (NGSEP, rice 15-kbp sample) to 0.79 (Hifiasm, rice 20-kbp sample). The number of m1 errors grew with the genome length for the assemblies generated by NGSEP and Flye. The number of m1 errors for the CHM13 assemblies of Canu and Hifiasm was lower than that of assemblies of plant samples generated by these tools. For the plant samples, the assemblies generated by Flye, NGSEP, and Canu reported similar numbers of total errors and each tool ranked first in one sample. Whereas Hifiasm plant assemblies ranked last, having between 1.38 and 2.2 times more misassemblies, the Hifiasm assembly of the CHM13 sample has the lowest number of errors. NGSEP ranked last for CHM13.

We also calculated measures of NA50 and NGA50, which estimate contiguity after structural inaccuracies are removed (Fig S1). Comparing NA50 and NGA50 values, we observed the same patterns described in the comparison of N50 and NG50 values. Contrasting NGA50 values with NG50 values, the NGSEP assemblies had the largest percentage of reduction for the rice samples (32%) and for CHM13 (20%; Table S1). In contrast, the Canu assemblies of the rice 20-kbp sample and the CHM13 preserved the same NG50 value as NGA50, regardless of the detected misassemblies. The Hifiasm assembly of the maize sample had the largest percentage of reduction (42%), followed by the Canu assembly (27%). The high number of errors observed in the Hifiasm assemblies of the rice samples translated into a reduction of 6 and 21% for the 15- and 20-kbp datasets, respectively. These numbers were lower than those of NGSEP and Flye, which suggests that the errors identified in the Hifiasm assemblies are located in small contigs, whereas errors in the other assemblies are more spread out across different contig lengths. Based on this measure, Canu ranked first in the rice 20-kbp sample, whereas Hifiasm ranked first in the other samples. NGSEP ranked third, and Flye ranked last in all samples.

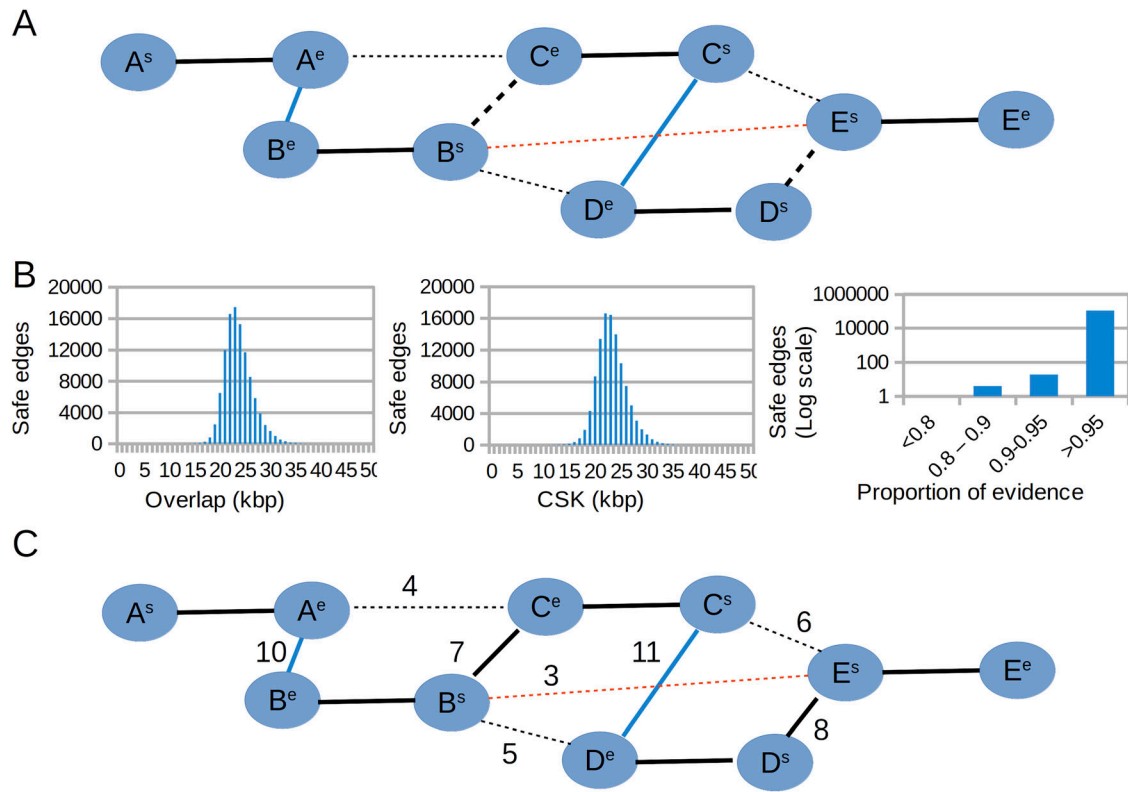

**Figure 2.  Layout algorithm.**
**(A)** Safe edges (blue) are selected as reciprocal best in both overlap and coverage of shared k-mers. The red edge represents a false positive. Bold solid black edges connect vertices of the same read. Bold dashed edges are true layout edges that are not reciprocal best. Other dashed lines represent true non-layout edges. **(B)** Distributions of overlap, coverage of shared k-mers, and proportion of evidence for safe edges of the rice 20-kbp PacBio HiFi data (details in the next section). **(C)** Log likelihoods are calculated for each edge based on the distributions; layout edges not selected in the first step are selected based on their ranking.

Regarding base pair–level quality, the number of mismatches and indel errors per 100 kbp was small in absolute numbers (less than 15 total errors per 100 kbp), which is a consequence of low base pair error rate of HiFi reads (Fig S2). The Flye assemblies consistently had the lowest total number of these errors, and in particular, they had up to three indel errors per 100 kbp. The greatest number of errors was observed in the CHM13 assemblies of NGSEP, Canu, and Hifiasm; in the Canu assembly of the rice 15-kbp dataset; and in the NGSEP assembly of maize. The number of mismatches in NGSEP assemblies seems to be correlated with the read depth of the samples.

Regarding computational efficiency, Fig 3C shows a comparison of the runtimes (having available 32 threads) required by each tool to assemble each of the datasets. Hifiasm and Canu are consistently the fastest and the slowest tools, respectively. NGSEP requires a lower runtime than Flye in all datasets except for the rice 15-kbp dataset, where Flye finishes 24 min faster than NGSEP. In absolute numbers, NGSEP is able to assemble the rice datasets in less than 4 h, the maize dataset in less than 8 h, and the CHM13 dataset in less than 18 h.

Combining the evaluation of accuracy and efficiency, NGSEP has better computational efficiency than Flye and Canu, and the assemblies have better contiguity than those of Flye and fewer

misassemblies than some of those assembled using Canu and Hifiasm.

## Assembly and haplotyping of diploid samples

We integrated our previous implementation of the ReFHap and the DGS algorithms to perform single-individual haplotyping of diploid heterozygous samples (Duitama et al, 2012). Unlike the previous implementation, which received a non-standard file with base calls for each heterozygous site, the two algorithms can now be executed from the VCF file with individual genotype calls and a BAM file with long reads aligned to the reference genome and sorted by reference coordinates. Moreover, we integrated the ReFHap algorithm within the assembly process of diploid samples to obtain phased genome assemblies from HiFi reads. ReFHap is executed independently on reads aligned to an initial assembly, which is generated using the methods described above for haploid samples. The goal of this phase is to identify and break edges in the assembly graph connecting reads sequenced from different haplotypes. Large deletions and regions of homozygosity larger than the read length usually break each contig into haplotype blocks (Cheng et al, 2021). Read depth within each block and between block boundaries is calculated to break the contig in contiguous regions classified as

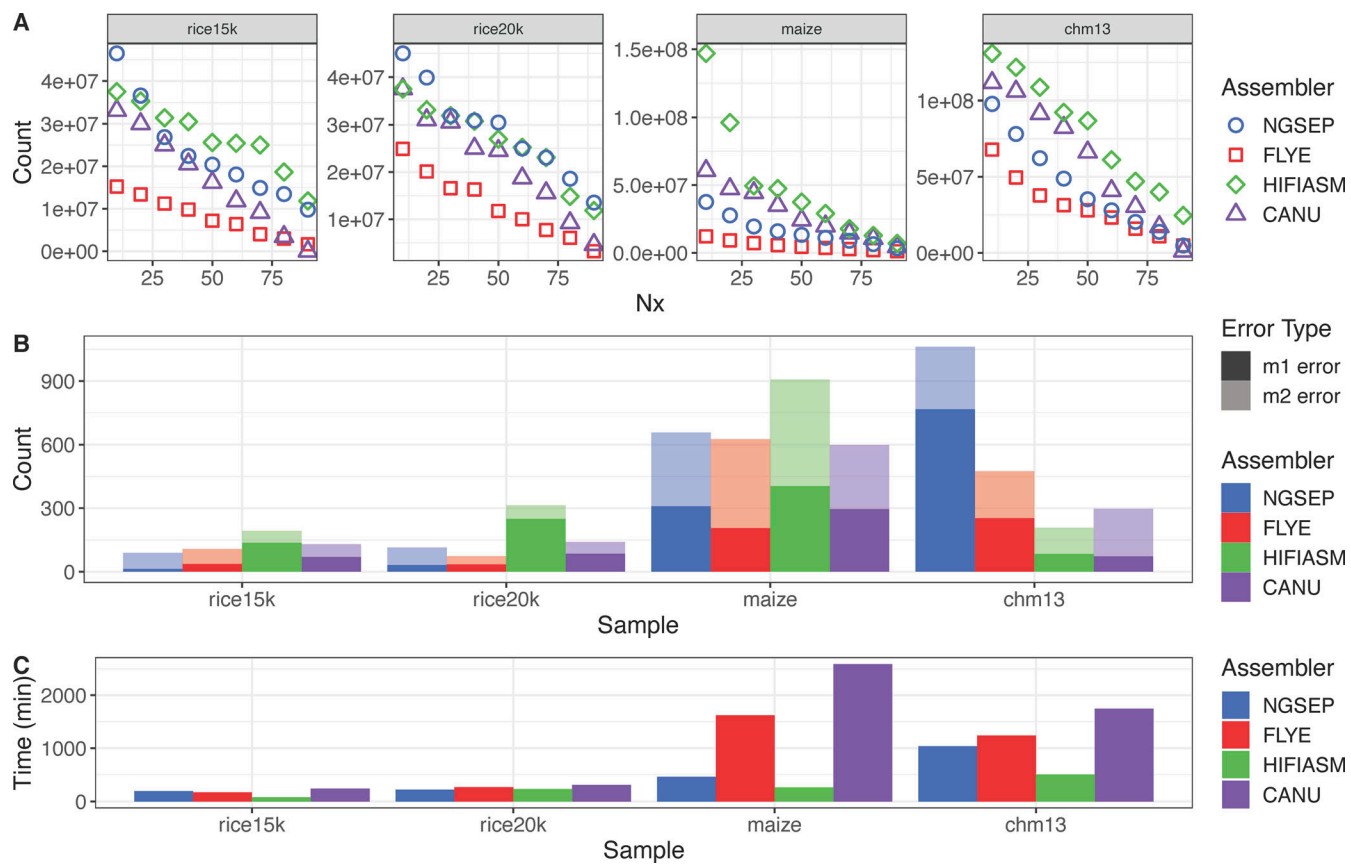

**Figure 3. Assembly results for haploid or inbred samples.**
**(A)** Nx curve. **(B)** Misassemblies (m1 error) and local misassemblies (m2 error) reported versus reference genomes. Rice15k corresponds to *Oryza sativa* 15-kbp HiFi reads, rice20k corresponds to *O. sativa* 20-kbp HiFi reads, maize corresponds to *Zea mays* B73 HiFi reads, and CHM13 corresponds to the human cell line CHM13 HiFi reads. **(C)** Execution time (in minutes) for each experiment.

true phased regions, large heterozygous deletions, or regions with high homozygosity. Edges connecting reads within true phased regions and assigned to different haplotype clusters are removed from the assembly graph.

To validate the accuracy of the complete process to assemble phased genomes, we first simulated two single-chromosome diploid genomes. The first was constructed from two publicly available MHC alleles. The second was constructed from the copies of the rice chromosome 9 corresponding to the Nipponbare and the MH63 assemblies. A high heterozygosity rate is expected in both cases. We assembled simulated reads from both individuals using both NGSEP and Hifiasm. For the MHC haplotypes, NGSEP was able to reconstruct the reference allele in two contigs of lengths 4.4 Mbp and 0.3 Mbp, and the alternative allele in three contigs of lengths 3.5, 0.5, and 0.2 Mbp (Fig S3). No switch errors (changes between real alleles within a contig) were detected in this assembly. Conversely, three contigs assembled by Hifiasm, with lengths of 4.4, 0.8, and 1.6 Mbp, mapped to the alternative MHC allele and one contig of 2.8 Mbp mapped to the reference MHC allele. Hence, the alternative allele was overrepresented, having the two smaller contigs embedded within the largest contig. The largest contig was also larger than the original allele because the left 100 kbp could not be mapped and the right 200 kbp was duplicated. In contrast, the

reference allele was subrepresented. Fig 4 shows the reconstruction of the rice alleles by NGSEP and Hifiasm. NGSEP assembled one large contig having three switch errors and five additional contigs covering the regions not covered by the first contig. Hifiasm assembled most of the MH63 chromosome in two contigs and most of the Nipponbare chromosome also in two contigs. Three switch errors were detected in this case. It also produced four small contigs (about 200 kbp), two of them overlapping with longer contigs.

To further assess the performance of NGSEP assembling diploid samples, we assembled publicly available HiFi reads of the human individual HG002. We obtained phased assemblies running NGSEP, Hifiasm, and Canu (Table S2). To compare statistics between phased assemblies and haploid assemblies merging the two haplotypes, we analyzed the primary assembly generated by Hifiasm and we also built a haploid assembly running Flye. NGSEP generated a phased assembly with a total length of 5.59 Gbp and an N50 of 1.1 Mbp, whereas Hifiasm and Canu produced phased assemblies of 5.98 and 6.16 Gbp, respectively. While the Hifiasm was highly contiguous (N50 = 67.02 Mbp), the N50 of the Canu assembly was 2.11 Mbp. This is consistent with the phasing strategy implemented in Canu, which generates a contiguous primary assembly and relatively small alternative haplotypes (Nurk et al, 2020). Regarding

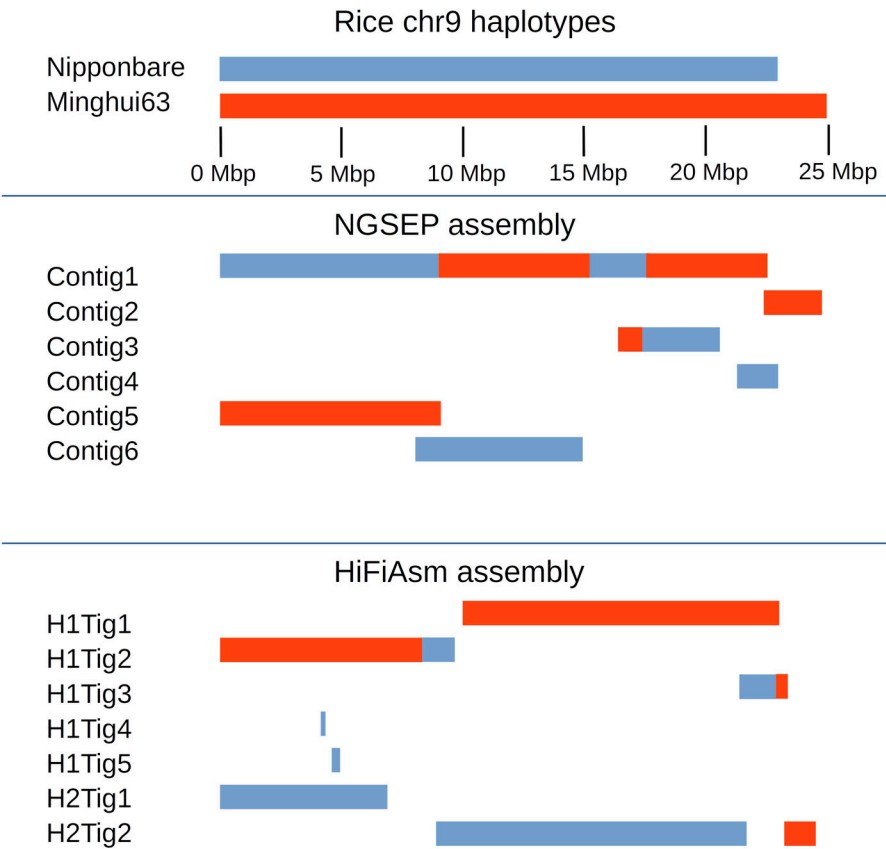

**Figure 4. Results of a diploid assembly of a simulated diploid individual built from the chromosome 9 sequences of the rice japonica accession Nipponbare and the indica accession Minghui63.**
Blue blocks show Nipponbare haplotypes, whereas red blocks indicate Minghui63 haplotypes. Changes in color in the same row represent switch errors.

haploid assemblies, whereas the total length of the primary assembly of Hifiasm was 3.11 Gbp, the length of the Flye assembly was 2.96 Gbp. The N50 of the primary assembly of Hifiasm was two times larger than the N50 of the Flye assembly (87.06 Mbp versus 45.03 Mbp).

To assess accuracy, we compared these assemblies with the current gold-standard phased assembly for this sample (Jarvis et al, 2022) running Quast (Gurevich et al, 2013). The three phased assemblies had a small reduction in NG50, relative to N50. The smaller total length of the NGSEP assembly produced genome coverage of 82.48%. This value was lower than the coverage of the Hifiasm assembly (94.46%) but higher than the coverage of the assembly generated running Canu (81.97%). The low genome coverage of the Canu assembly can be explained because the total length of contigs larger than 50 kbp is reduced to 5.53 Gbp. As expected, the genome coverage of the haploid assemblies was around 50%. Regarding errors, according to Quast, the ranking among phased assemblies in almost all error types was first, Canu; second, Hifiasm; and third, NGSEP. The two exceptions were the number of indels per 100 kbp, in which Hifiasm ranked first, and the number of m2 errors, in which Hifiasm ranked last. In absolute values, the number of m1 errors ranged between 1,500 (Canu) and 2,591 (NGSEP) and the number of m2 errors ranged between 1,092 (Canu) and 1,480 (Hifiasm). The

numbers of errors were between 3 and 29 times larger than those observed for CHM13. M1 and m2 errors for haploid assemblies were lower than errors in phased assemblies, except for m2 errors observed in the Flye assembly, but the values were within the same order of magnitude. The number of mismatch errors per 100 kbp was about twice the number observed in the CHM13 sample, and the number of indels per 100 kbp grew up to seven times in proportion (~20 indels per 100 kbp). Combining contiguity with accuracy, although the best NGA50 was obtained by Hifiasm (5.55 Mbp), this value is only 8.9% of the original NG50. Canu and NGSEP had similar NGA50 (0.925 and 0.871 Mbp, respectively).

Understanding that in this comparison some of the misassembly errors could be due to switch errors between the gold-standard haplotypes, we estimated the number of switch errors by two different methods based on parental-specific k-mers. First, we ran the algorithm implemented in Merqury (Rhie et al, 2020). For phased assemblies, the number of switch errors ranged from 41,818 (Canu) to 66,230 (Hifiasm). In contrast, highly diverging values were observed in the haploid assemblies (Hifiasm: 33,872; Flye: 1,749,585). The switch error rate was around 0.1%, except in the Flye assembly (6.09%). However, in absolute numbers, switch errors were much larger than misassembly errors reported by Quast. Hence, we decided to implement an alternative algorithm to infer switch

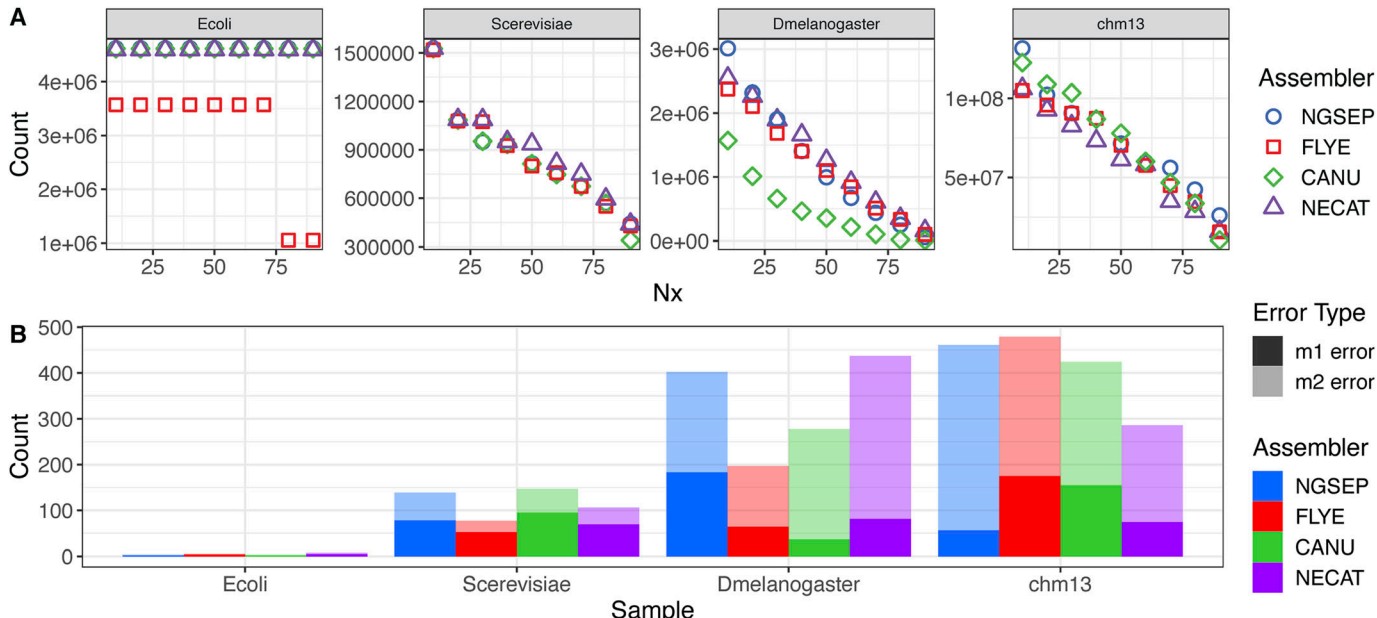

**Figure 5. Haploid genome assembly results using ONT reads.**
**(A)** Nx curve. **(B)** Misassemblies (m1 error) and local misassemblies (m2 error) reported for each genome versus reference genomes.

errors from parental-specific k-mers (see the Materials and Methods section for details). With this method, the number of predicted switch errors was reduced to a range between 1,146 and 3,807, which is more comparable with the number of misassembly errors. Comparing genome assemblies, the tool ranking was the same as that derived from the number of switch errors reported by Merqury.

We also collected some of the metrics proposed by Cheng et al (2021), related to the ability of the assembly to reconstruct the two alleles of each gene present in the diploid sample. Over 98% of the 18,558 single-copy protein-coding genes in the reference genome were recovered at least one time in all assemblies (phased and haploid). However, two or more alleles were mapped for 17,679 genes (95.26%) in the Hifiasm assembly, whereas this number was 14,294 (77.02%) for the NGSEP assembly, and 14,100 (75.98%) for the Canu assembly. As expected, the percentages for the haploid assemblies were below 0.5%. From the remaining 1,386 multicopy genes in the reference genome, more than two alleles were mapped for 1,236 genes (89.18%) in the Hifiasm assembly, for 1,002 genes (72.29%) in the NGSEP assembly, and for 1,219 genes (87.95%) in the Canu assembly. These numbers are consistent with the observed genome coverage. In terms of computational efficiency, NGSEP and Hifiasm were able to reconstruct the genome in about 50 h using 32 threads, whereas Flye took 79 h and Canu took 158 h.

### Benchmark with ONT data

To test the performance of our algorithms with Nanopore reads, we downloaded and assembled datasets of Nanopore reads sequenced from samples of *E. coli, Saccharomyces cerevisiae, D. melanogaster*, and the CHM13 human cell line. We compared the assemblies obtained using Canu, Flye, and NECAT, and the assembly

obtained by NGSEP on the reads corrected by NECAT. Fig 5 shows the statistics of these assemblies comparing these tools. Complete assembly statistics are shown in Table S3. For *E. coli* and *S. cerevisiae*, all tools generated nearly perfect assemblies in terms of contiguity, except Flye, which reported two contigs for the *E. coli* sample (Fig 5A). In the *E. coli* sample, the misassemblies were negligible and the NGA50 of 3.3 Mbp is likely produced by disagreement on the start of the circular chromosome. On the yeast assemblies, between 100 and 200 misassemblies identified consistently reduce the NGA50 to about 400 kbp.

The fruit fly genome assemblies produced by the different tools were more divergent and contained the largest differences between N50 and NG50 across the datasets (Fig S4). NECAT produced an assembly of 251 contigs with an N50 of 1.36 Mbp. However, the total assembly length was only 126 Mbp (143 Mbp expected), resulting in a reduced NG50 of 1.09 Mbp. The Flye assembly was similar, having a total length of 128 Mbp and an NG50 of 1.11 Mbp. In contrast, Canu reported 577 contigs with an N50 of 0.48 Mbp and an NG50 of 0.45 Mbp. The NGSEP assembly had an N50 of 1.05 Mbp but a total length of 139 Mbp, which was the closest to the expected length. As a result, the NG50 was only reduced to 1.0 Mbp. Regarding misassembly errors, the NGSEP assembly had the largest number of m1 errors and Canu had the largest number of total errors (Fig 5B). As a consequence, the Flye assembly retained an NGA50 value of 1.01 Mbp, whereas the NGA50 value of the NGSEP assembly was reduced to 0.66 Mbp, slightly larger than the NGA50 of NECAT.

Regarding base pair errors, the values were in general one order of magnitude larger than those obtained from HiFi reads (Fig S5). Only the mismatch errors of the *E. coli* assemblies were comparable to those obtained using HiFi reads. The Flye assemblies had the smallest numbers of errors, and the Canu assemblies had the largest number of errors. The NGSEP and NECAT assemblies had a

similar intermediate behavior, with NGSEP making more mismatch errors and NECAT making more indel errors.

Finally, we assembled ultralong Nanopore reads (average over 200 kbp) sequenced from the human haploid cell line CHM13. Unfortunately, Flye and Canu failed to assemble these sequences with the available computational resources, requiring more than 512 GB of memory for this process. Hence, we compared our results with publicly available assemblies (https://github.com/marbl/CHM13/blob/master/Sequencing_data.md corresponding to rel3) generated using these tools. The Canu assembly had the largest N50 value (78 Mbp), and the NECAT assembly had the lowest value (61 Mbp). The Canu assembly also had the largest total assembly length (2.97 Gbp), which was also the closest to the expected length of 3.12 Gbp. However, the Canu and the Flye assemblies had between two and three times more m1 errors, compared with the NGSEP and the NECAT assemblies. Although the NGSEP assembly had the largest number of m2 errors, the resulting NGA50 of NGSEP was the highest, followed by Canu (Fig S4). The NGSEP and the NECAT assemblies also had a small number of base pair errors (about 10 mismatches and 50 indels per 100 kbp) compared with the assemblies of Flye and Canu, which had over 300 errors per 100 kbp.

### Other related features

Based on the development of the genome assembler, version 4 of NGSEP also includes a module to calculate the spectrum of k-mer counts, either from sequencing reads or from a genome assembly. For a k-mer size less or equal to 15, the k-mer counts are stored in a fixed array of 2-byte integers of size $2^{30}$. This allows to create the spectrum with a fixed RAM usage of 2 gigabytes for an arbitrary number of input reads. Based on this spectrum of k-mers, we included a functionality for error correction in which substitution errors can be corrected by looking at single changes producing k-mers within the distribution of k-mer counts. Moreover, the minimizer table generated to perform efficient identification of read overlaps was also used to create a reference alignment tool for long reads. To keep the algorithm memory tractable, minimizers appearing 1,000 or more times within the reference sequence are discarded. Minimizers for each read are calculated and searched in the minimizer table corresponding to the reference sequence. Minimizer hits are interpreted as k-mer ungapped alignments and clustered according to the read start site predicted for each read. We assessed the performance of the minimizer algorithm implemented in NGSEP for aligning simulated long reads, comparing the results with the alignments obtained using Minimap2 (Li, 2018). Both tools achieved almost perfect accuracy for *Saccharomyces aureus* and *S. cerevisiae* genomes. Minimap2 showed 3% higher mapping accuracy for the experiment with the human chr20, but NGSEP reported lower root mean squared error values (Fig S6).

Finally, for circular genomes we implemented a circularization feature as an option of the genome assembler. Given an input set of possible origin sequences, NGSEP maps these sequences to the assembled contigs using the long-read alignment algorithm. Each presumably circular contig is rotated and oriented based on the best alignment of an origin sequence.

## Discussion

In this work, we present the results of our latest developments to facilitate de novo construction of genome assemblies using long reads, which includes novel algorithmic approaches to perform the different steps of the overlap–layout–consensus model. In summary, we formalize the construction of an undirected graph to handle the different types of overlaps taking into account the sequencing strand of each read. For the graph construction step, we propose an informed hash function based on the distribution of k-mer counts to improve the selection of minimizers. We also propose a new algorithm to build layouts from our assembly graph in which layout edges are selected based on a Bayesian scoring function. Finally, we integrated previous works on haplotyping to filter edges in the assembly graph and to build phased assemblies. Experiments with a wide variety of datasets indicate that our approach achieves competitive accuracy and efficiency, compared with state-of-the-art tools. From the user perspective, NGSEP achieves nearly perfect assemblies for several species and it is able to reconstruct most gene-rich regions, even in complex genomes. One major advantage of our software is that, combined with previous developments, it offers an easy-to-use, open-source, and platform-independent framework to run a complete analysis of high-throughput sequencing reads, including de novo assembly, read mapping, variant detection, genotyping, and downstream analysis of genomic variation datasets.

The algorithms designed and implemented in NGSEP contribute new alternatives to identify solutions to the genome assembly problem. Although the graph construction with two vertices per read has been used in previous works (Miller et al, 2008; Koren et al, 2017), current software tools seem to implement the classical directed string graph, which requires taking early decisions on the orientation of each read (Cheng et al, 2021). We believe that the undirected graph used in this work makes a better representation of DNA sequences compared with the string graph because it takes into account that DNA is double-stranded, and hence, it captures more information from the input reads. This allows devising algorithmic approaches different from a greedy traversal of a curated string graph. Moreover, to achieve improved computational efficiency, we avoided complete alignments between reads. Instead, we performed estimations of different types of information (overlap, CSK, and percentage of the overlap supported by evidence), which can be used as features to select edges building assembly paths based on a likelihood calculation for each edge. The layout algorithm of NGSEP is inspired by the classical Christofides algorithm for the travel salesman problem, treating the path construction as an edge selection process. Edge features are combined based on their likelihood, replacing edge filtering with edge prioritization. This approach eliminates the need of hard filtering decisions and makes the algorithm adaptable to genomic regions with different repeat structures and to the analysis of reads with variable sequencing error rates.

Taking Nx curves and misassemblies into consideration, NGSEP produces high-quality assemblies with higher contiguity than Flye and, in some cases, a lower number of errors compared with Canu and Hifiasm. These statistics suggest that NGSEP can be used as an

accurate alternative to assemble PacBio HiFi reads. Although further work is required to improve N50 in complex assemblies (especially human diploid samples), our results indicate that the NGSEP assemblies are able to reconstruct the two copies for over 70% of the genes in the human genome. Recent works indicate that assembling phased complex genomes with chromosome-level accuracy still requires the integration of data from technologies or strategies that provide scaffolding and phasing information such as Hi-C or parental sequencing (Garg et al, 2021; Kronenberg et al, 2021; Porubsky et al, 2021; Cheng et al, 2022; Jarvis et al, 2022; Nurk et al, 2022). We plan to implement alternatives to integrate these data in future releases of NGSEP. However, our experiments with diploid samples indicate that new algorithms implemented in existing or novel tools could significantly improve the accuracy of phased assemblies directly from long reads.

Regarding Oxford Nanopore reads with high error rates, NGSEP was able to perform accurate assemblies after reads were corrected by running the specialized algorithm implemented in NECAT. This error correction step is crucial in the assembly process of current ONT reads. However, upcoming improvements in the read quality are likely to produce ONT HiFi reads, eliminating the need of a specialized error correction step. The highly contiguous and structurally correct assembly of the CHM13 sample obtained from ultralong ONT reads suggests that this is a promising alternative to achieve chromosome-level assemblies. Improvements in base pair quality are needed not only to improve the base pair accuracy, but also to identify heterozygous variants accurately and to achieve phased assemblies, as it can be done using HiFi reads.

We believe that the new algorithms presented in this study make a significant contribution to the development of bioinformatic algorithms and tools for genome assembly. Moreover, the new functionalities of NGSEP facilitate the construction of genome assemblies for researchers working on a wide range of species.

# Materials and Methods

## Benchmark datasets

PacBio and Nanopore publicly available raw datasets were retrieved from NCBI. Haploid datasets included PacBio HiFi/circular consensus sequence 20-kbp reads from the *Oryza sativa* indica MH63 accession (PRJNA558396) (Song et al, 2021), 15-kbp reads from the *O. sativa* indica MH63 accession (SRR10188372), the *Zea mays* B73 accession (PRJNA627939) (Hon et al, 2020), and the CHM13 human haploid cell line (PRJNA530776) (Nurk et al, 2022). As a diploid benchmark dataset, we downloaded reads from the human male HG002/NA24385 (PRJNA586863). Nanopore reads for *E. coli* K12 were obtained from the Loman Lab available at http://lab.loman.net/2015/09/24/first-sqk-map-006-experiment/ (Loman et al, 2015). We selected run MAP-006-1, which also corresponds to the dataset used by Canu in their tutorial. Nanopore reads for *S. cerevisiae* and *D. melanogaster* were directly downloaded from http://www.tgsbioinformatics.com/necat/ (Chen et al, 2021). Nanopore reads for the human cell line CHM13 corresponding to

the release 8 (rel8) were downloaded from https://github.com/marbl/CHM13/blob/master/Sequencing_data.md

## Comparison of long-read haploid genome assembly tools

We compared the performance of the algorithm described in this work with the algorithms implemented in HiCanu (Nurk et al, 2020), Flye (Kolmogorov et al, 2019), and Hifiasm (Cheng et al, 2021) for PacBio HiFi reads; and with the algorithms implemented in Canu, Flye, and NECAT (Chen et al, 2021) for Nanopore reads. WTDBG (Ruan & Li, 2019) was not included because in some initial benchmark experiments, it reported a much lower accuracy for complex genomes, compared with other tools, and because it seems to be replaced by Hifiasm. All PacBio assemblies were run in a Microsoft Azure Standard E64as_v4 (64 vCPUs, 512 GiB memory) virtual machine. The parameters used for each tool are detailed in Tables S4 and S5.

## Comparison of genome assemblies with reference genomes

To compare the assembly achieved by each tool against a reference genome, we used Quast (Gurevich et al, 2013) with default parameters for the *E. coli* and *S. cerevisiae* samples, and with the following parameters for genomes larger than 100 Mb: --eukaryote --min-contig 25,000 --min-identity 99 (98 for ONT reads) --min-alignment 5,000 --extensive-mis-size 20,000. Whereas reference coverage, assembly length, and N50 were used as sensitivity measures, the number and type of misassemblies were used as specificity measures. We calculated and compared these statistics among all assemblies per dataset. The Nx curve was also calculated for each assembly. The reference genomes used in the comparison were *O. sativa* indica MH63 (CP054676–CP054688) (Song et al, 2021), *Z. mays* B73 v.5 (GCA_902167145.1) (Jiao et al, 2017), the recently published telomere-to-telomere assembly of the human haploid line CHM13 v2.0 (GCA_009914755.4) (Nurk et al, 2022), and the genomes of *D. melanogaster* v.6, *E. coli* K12, and *S. cerevisiae* S288c, which were downloaded from the NECAT web site (Chen et al, 2021).

## Diploid genome benchmarking

Simulations: To assess the accuracy of the algorithm implemented in NGSEP for reconstruction of diploid samples, we simulated two single-chromosome individuals. First, we built a synthetic individual joining two different MHC alleles: the reference allele extracted from GRCh38, and an alternative reconstruction available at the NCBI nucleotide database (accession NT_167249), generated as part of the MHC haplotype project (Horton et al, 2008). Second, we built an individual joining the rice chromosome 9 reconstructions of the reference genome (Nipponbare) and MH63. We simulated 10,000 and 125,000 reads, respectively, from each simulated diploid individual using the SingleReadsSimulator of NGSEP with an average length of 20 kbp, an SD of 5 kbp, a substitution error rate of 0.5%, and an indel error rate of 1%.

HG002: NGSEP v4.3.1. Hifiasm v0.16.0 (Cheng et al, 2021), Canu 2.1.1 (Nurk et al, 2020), and Flye v2.8.3 (Kolmogorov et al, 2019) were executed to obtain phased and haploid assemblies for the Personal Genome Project Ashkenazi Jewish son HG002 (four runs with

accession numbers SRR10382244, SRR10382245, SRR10382248, and SRR10382249). We registered execution time over a node with an AMD EPYC 7402 2.80 Hz, 24C/48T, 128M Cache, a DDR4-3200 processor, 32 cores, and 512 Gb of RAM. We converted the output files from HiFiAsm (*ctg.gfa) to fasta (*.fa) and merged the haplotypes (*hap1.p_ctg.fa and *hap2.p_ctg.fa) to calculate the main metrics and compare against the NGSEP diploid assembly. We ran Quast v5.0.2 (Gurevich et al, 2013) to assess the accuracy of the assemblies, using as reference the HG002 diploid assembly reported by Jarvis et al (2022). Because structural errors reported by Quast can really be the product of switch errors, we calculated the number and proportion of switch errors using the k-mer–based method implemented in Merqury (Rhie et al, 2020). We also implemented a script available in NGSEP (class ngsep.benchmark.KmerBasedSwitchErrorsFinder), which divides the genome in overlapping windows of 20 kbp (10-kbp overlap), assigns windows to parental haplotypes based on parental-specific k-mers, and identifies neighbor windows providing evidence of switch errors.

We also calculated the protein-coding genes that can be identified in the assemblies and the number of alleles identified for each gene, following a procedure similar to that explained in Cheng et al (2021). We mapped the main transcripts of 19,944 protein-coding genes to each assembly using Minimap2 (Li, 2018). After sorting alignment (.paf) files by query name, we built a script within NGSEP to calculate the number of alleles identified for each gene (class ngsep.benchmark.AssembliesAligned-TranscriptStatistics). Results of this script are available in Table S2. Assuming that phased assemblies should have two copies of unique genes and more than two copies of genes with paralogs, gene mapping can be used to evaluate the completeness of phased genome assemblies. A gene is considered as a single copy (SC) if only one match is identified in the reference genome (at 99% identity). Otherwise, it is considered a multicopy (MC) gene.

### Accuracy assessment for long-read alignment

Simulated reads were aligned against their respective reference sequence using Minimap2 v2.17 (Li, 2018) and the ReadsAligner command of NGSEP v4.2.1 with k-mer lengths of 15 (default mode) and 20. Default parameters were used for all aligners. For time performance evaluation, we conducted all alignments using four cores of processing and 20 GB of memory. We evaluated the accuracy of the aligners using the percentage of aligned reads, and sensitivity and false-positive rate metrics. These metrics were calculated using a script that, taking an alignment file as input, infers the real position in the reference genome for each aligned read from the read name and calculates the difference with the position where the read is aligned. Total alignment rate and root mean squared error are calculated after the total number of aligned reads is counted and the square error rate is totalized over the alignments. This script is available with the NGSEP distribution (class ngsep.benchmark.QualityStatisticsAlignmentSimulatedReads). Accuracy metrics were computed for bam files filtered by alignment quality values from 0 to 80.

## Data Availability

The software described in this study is available as part of the open-source software product NGSEP. Stable versions are available in SourceForge (http://ngsep.sf.net). The version under development is available on GitHub (https://github.com/NGSEP). All the experiments presented in this study were performed using previously available public data from different repositories. The Materials and Methods section includes the specific database and accession id for each dataset and each reference genome. Assemblies generated using NGSEP v4.3.1 are available in SourceForge (https://sourceforge.net/projects/ngsep/files/benchmarkAssembler/).

## Supplementary Information

## Acknowledgements

This work was supported by the Colombian Ministry of Sciences research fund "Patrimonio Autónomo Fondo Nacional de Financiamiento Para la Ciencia, la Tecnología Y la Innovación Francisco José de Caldas" through the grant with contract number 80740-441-2020, awarded to J Duitama. This work was also supported by internal funds of Universidad de Los Andes through the FAPA initiative and through the publication fund led by the Vice Presidency of Research and Knowledge Creation. We also wish to acknowledge the support of the IT Services Department and ExaCore—IT Core-facility of the Vice Presidency for Research & Creation at the Universidad de Los Andes that allow us to perform the computational analysis.

### Author Contributions

L Gonzalez-Garcia: software, formal analysis, validation, investigation, visualization, methodology, and writing—original draft, review, and editing.
D Guevara-Barrientos: software, formal analysis, validation, investigation, and writing—review and editing.
D Lozano-Arce: formal analysis, validation, investigation, and writing—review and editing.
J Gil: validation, investigation, visualization, methodology, and writing—original draft, review, and editing.
J Díaz-Riaño: validation, investigation, visualization, methodology, and writing—original draft, review, and editing.
E Duarte: software, validation, investigation, and writing—review and editing.
G Andrade: software, validation, investigation, methodology, and writing—review and editing.
JC Bojacá: software, validation, investigation, methodology, and writing—review and editing.
MC Hoyos-Sanchez: formal analysis, validation, investigation, and writing—review and editing.
C Chavarro: software, validation, investigation, methodology, and writing—review and editing.

N Guayazan: software, validation, investigation, methodology, and writing—review and editing.

LA Chica: formal analysis, validation, investigation, visualization, and writing—review and editing.

MC Buitrago Acosta: formal analysis, validation, investigation, and writing—review and editing.

E Bautista: formal analysis, validation, investigation, and writing—review and editing.

M Trujillo: software, formal analysis, validation, investigation, methodology, and writing—review and editing.

J Duitama: conceptualization, software, formal analysis, supervision, funding acquisition, validation, investigation, visualization, methodology, project administration, and writing—original draft, review, and editing.

## Conflict of Interest Statement

The authors declare that they have no conflict of interest.

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
