## [Reviewer comments · Life Science Alliance]

Life Science Alliance

New algorithms for accurate and efficient de-novo genome assembly from long DNA sequencing reads

Laura Gonzalez-Garcia, David Guevara-Barrientos, Daniela Lozano-Arce, Juanita Gil, Jorge Diaz-Riano, Erick Duarte, German Andrade, Juan Bojaca, Maria Hoyos-Sanchez, Christian Chavarro, Natalia Guayazan, Luis Chica, Maria Buitrago Acosta, Edwin Bautista, Miller Trujillo, and Jorge Duitama

DOI: <https://doi.org/10.26508/lsa.202201719>

Corresponding author(s): *Jorge Duitama, Universidad de Los Andes*

Review Timeline:

Submission Date:	2022-09-10
Editorial Decision:	2022-10-18
Revision Received:	2023-01-11
Editorial Decision:	2023-02-09
Revision Received:	2023-02-10
Accepted:	2023-02-13

Transaction Report:

October 18, 2022

Re: Life Science Alliance manuscript #LSA-2022-01719-T

Jorge Duitama

Dear Dr. Duitama,

Thank you for submitting your manuscript entitled "New algorithms for accurate and efficient de-novo genome assembly from long DNA sequencing reads" to Life Science Alliance. The manuscript was assessed by expert reviewers, whose comments are appended to this letter. We invite you to submit a revised manuscript addressing the Reviewer comments.

Thank you for this interesting contribution to Life Science Alliance. We are looking forward to receiving your revised manuscript.

Sincerely,

B. MANUSCRIPT ORGANIZATION AND FORMATTING:

Reviewer #1 (Comments to the Authors (Required)):

General comments:

Gonzalez-Garcia generate a new contig generating algorithm for genome assembly with principled improvements over past algorithms, in employing an undirected graph that considers reads from both strands of the DNA molecule, and has a haplotype phasing mode for more accurate diploid assemblies. They compare their contig algorithm against some of the state of the art algorithms on 4 species, from plants to humans. Their metrics are either on par, a little worse, or better than the other state of the art algorithms depending on species. I think it is worth publishing this study, and getting the word out about it.

However, before doing so, I think it needs more work, both in terms of manuscript preparation and more analyses. The authors don't mention or compare their results to the some more advanced algorithms available, and don't cite some of key papers from 2021 and earlier this year in 2022. I think they should include a comparison with HiFiasm Hi-C for haplotype phasing (Cheng et al 2022 Nature Biotech). If the authors have difficulty implementing it, a user friendly version has been installed in the Galaxy platform, and can be used for free within the Vertebrate Genomes Project framework.

Specific comments:

Line 66, the 30Kb apparently is referring to Pacbio CLR reads. The CCS reads are 15-20Kb.

Line 69. I believe the first version of the CLR reads began with a 15% error rate. But Pacbio improved the technology to where the error I believe was below 10%.

Line 73. Should cite Koren et al 2018, Nature Biotechnology for the long read trio approach.

The authors should also cite here and elsewhere for the appropriate comparisons the human pangenome bakeoff study on HG002 (Jarvis et al 2022 bioRxiv). Besides comparing all the major algorithms (including HiCanu, HiFiasm, Canu, Flye, etc), a major conclusion is that the highest quality assemblies are achieved when including phasing information during the assembly graph stage, instead of before or after. This is what HiFiasm Trio and now HiFiasm Hi-C does. It would be good to compare the phasing version of NGSEP with these approaches, as a more fair comparison. A revised version of the this study will be published in Nature on Oct 19th.

The authors should also cite the VGP assembly paper for some of the discoveries mentioned in the introduction, especially the importance of haplotype phasing (Rhie et al 2022 Nature).

Line 129. Is the fixed array $2 > 30$ due to compute limitations, or theoretical sequence k-mer limitations of what is represented in DNA? It is not clear why the k-mer size can't be bigger than 15 nucleotides in length.

Line 155. Indel errors are not entirely random across the genome. They occur more often in homopolymer regions, especially with long reads.

Line 206. What reads were used for polishing? The HiFi reads, or Illumina reads? If short reads, they are no longer necessary for polishing HiFi based contigs.

Also in this section of the paper, should mention that these assemblies are only contigs, not scaffolds.

Need to mention that CHM13 is not truly a diploid genome; it is effectively haploid, making the assembly process easier. Also need to say that the reference being compared is the T2T complete CHM13 genome.

Line 423: Should cite Cheng et al 2022 Nature Biotech also for HiFiasm Hi-C, and Kronenberg et al 2021 Nature Communications for the first algorithm that used Hi-C for haplotype phasing.

Line 446: Cite Jarvis et al 2022 bioRxiv for the source of the HG002 data. It might be good to also used the genome in this study as the highest quality HG002 reference at this point in time.

Reviewer #2 (Comments to the Authors (Required)):

The manuscript describes new algorithms for long-read assembly, which are implemented as a part of the existing NGSEP package. De novo assembly is indeed a key algorithmic challenge in computational biology and any improvements in this field are highly appreciated. The authors benchmark their approach against the state-of-the-art methods using a variety of PacBio and Oxford Nanopore datasets. On most datasets, NGSEP is an average performer in the most important contiguity metrics. It is nevertheless an interesting orthogonal approach with a few algorithmic novelties and will certainly have its use cases. I am enthusiastic about the manuscript in general, but have a few important requests to ensure that the algorithmic novelties and key comparisons are presented clearly.

1. In the description of the algorithms, the authors provide a fair amount of technical detail. It makes it somewhat difficult to understand what are key algorithmic novelties, compared to the existing assemblers. I encourage the authors to provide shorter summaries for non-expert readers that highlight the key differences compared to the other approaches, and the best use case scenarios.

2. The N50 metric depends on the assembly size and may be inflated for assemblies with lower genome coverage. It should be substituted with the NG50 metric which does not depend on the assembly size.

3. In addition, I encourage the authors to provide the NGA50 metric, which accounts for possible structural inaccuracies. The authors report NGA50 values in the supplementary tables, but they should be mentioned in the main text (and perhaps added to the NG50 plots).

4. According to the methods section, the authors ran QUAST with default parameters, which are optimized for bacterial genomes. The large genome mode "--large" should be used instead for all datasets. This mode accounts for possible structural differences between reference and assembled genomes (as mentioned by the authors) and may substantially affect misassemblies counts, NGA50 and other alignment-based statistics.

5. The main text (and perhaps figures) should also contain information about the base-level assembly quality (number of mismatches and small indels).

6. The authors rely on the misassemblies count to argue that their approach is more accurate, however this metric could be misleading. Misassemblies are often clustered together inside difficult-to-assemble regions (e.g. segmental duplications), and reflect the differences between reference and assembled genomes or alignment difficulties. As a result, assemblies that do not attempt to reconstruct such regions will have less reported misassemblies without significant penalties in completeness metrics. One important change to alleviate this is to use the correct QUAST parameters (as mentioned above). A good strategy is to mask the difficult-to-profile regions of the genome (see the Shasta paper for example), but this may be difficult to do for non-human genomes. An alternative strategy may be to focus only on very large misassemblies (10, 20, 50kb+). The minimum misassembly size should be reported in the main text.

7. For the diploid assembly evaluations, the authors should report hamming and switch errors, since the usual contiguity metrics are not sensitive to phasing errors.

8. NGA50 for the phased HG002 was much lower (1.68 Mb), in comparison to the primary hifiasm mode (69 Mb). However I would expect the phased hifiasm mode to have much lower contiguity than currently reported. Did the authors run hifiasm using only HiFi reads, or was this assembly generated with the help of HiC/trio? The comparison with HiFi only phased assembly would be more relevant.

9. The authors tested their approach on nanopore data using three relatively small datasets. I suggest providing results for human genomes (CHM13, HG002), for which a lot of data is publicly available. Modern nanopore reads are much longer than PacBio HiFi and should result in longer phased blocks, highlighting a good use case for the authors' method.

10. The authors need to provide a link to a publicly available repository with the source code and instructions how to run the assembler and reproduce the paper analysis. All assemblies generated in this study should be also made publicly available.

Reviewer #1 (Comments to the Authors (Required)):

General comments:

Gonzalez-Garcia generate a new contig generating algorithm for genome assembly with principled improvements over past algorithms, in employing an undirected graph that considers reads from both strands of the DNA molecule, and has a haplotype phasing mode for more accurate diploid assemblies. They compare their contig algorithm against some of the state of the art algorithms on 4 species, from plants to humans. Their metrics are either on par, a little worse, or better than the other state of the art algorithms depending on species. I think it is worth publishing this study, and getting the word out about it.

However, before doing so, I think it needs more work, both in terms of manuscript preparation and more analyses. The authors don't mention or compare their results to the some more advanced algorithms available, and don't cite some of key papers from 2021 and earlier this year in 2022. I think they should include a comparison with HiFiasm Hi-C for haplotype phasing (Cheng et al 2022 Nature Biotech). If the authors have difficulty implementing it, a user friendly version has been installed in the Galaxy platform, and can be used for free within the Vertebrate Genomes Project framework.

R. We thank the reviewer for the assessment of our work. We improved the benchmark experiments and updated the text to address properly each comment included in the review. Following the recommendations of both reviewers, we improved the assessment of the phased assembly of HG002, using as gold standard the phased assembly published by Jarvis et al. 2022 and expanding the tools included in the comparison and the evaluation procedures. We also improved the literature review following the suggestions of the reviewer. Please see below our specific answer to each comment.

Specific comments:

Line 66, the 30Kb apparently is referring to Pacbio CLR reads. The CCS reads are 15-20Kb.

R. The reviewer is right. We refer to the CLR protocol. We adjusted the text to make this clear

Line 69. I believe the first version of the CLR reads began with 15% error rate. But Pacbio improved the technology to where the error I believe was below 10%.

R. We adjusted the text to clarify that the 15% error rate was correct only for early releases of both technologies.

Line 73. Should cite Koren et al 2018, Nature Biotechnology for the long read trio approach.

R. We cited Koren et al. 2018 at this point and moved Wenger et al. 2019 to a more appropriate part of the manuscript.

The authors should also cite here and elsewhere for the appropriate comparisons the human pangenome bakeoff study on HG002 (Jarvis et al 2022 bioRxiv). Besides comparing all the major algorithms (including HiCanu, HiFiasm, Canu, Flye, etc), a major conclusion is that the highest quality assemblies are achieved when including phasing information during the assembly graph stage, instead of before or after. This is what HiFiasm Trio and now HiFiasm Hi-C does. It would be good to compare the phasing version of NGSEP with these approaches, as a more fair comparison. A revised version of this study will be published in Nature on Oct 19th.

R. We cited appropriately the work of Jarvis et al., 2022, as recently published in Nature. Moreover, we used the HG002 genome as a gold standard to improve the benchmarking results in diploid individuals. We wish to clarify though that the current version of our assembler does not include Hi-C reads. Hence, with the goal of comparing the algorithms based on the same input data (only HiFi reads) and following the advice of the second reviewer, we performed the comparison against the assembly obtained by HiFiAsm using only HiFi reads. If the goal would be to produce the best possible phased genome assembly, we agree with the reviewer and with Jarvis et al., 2022 that integrating Hi-C data during the graph construction is currently the best way to achieve this goal. We plan to integrate parental and Hi-C data as future developments of our genome assembler. We improved the discussion to clarify these points.

The authors should also cite the VGP assembly paper for some of the discoveries mentioned in the introduction, especially the importance of haplotype phasing (Rhie et al 2022 Nature).

R. We cited the VGP paper (Rhie et al. 2021) in the first paragraph of the introduction

Line 129. Is the fixed array $2^{>30}$ due to compute limitations, or theoretical sequence k-mer limitations of what is represented in DNA? It is not clear why the k-mer size can't be bigger than 15 nucleotides in length.

R. The limit is related to the maximum number that can be used as an array index in java. The limit of 2^{30} allows to perform direct hashing of DNA 15-mers to calculate k-mer counts. In principle we could increase the k-mer length to 16 or 17 by using two / four integer arrays respectively. However, the code becomes much more complicated and we believe that the improvement would be marginal.

Line 155. Indel errors are not entirely random across the genome. They occur more often in homopolymer regions, especially with long reads.

R. We improved the wording of this sentence because we really did not want to say that the errors are randomly distributed across the sequence (and we actually do not assume that in our method). We agree with the reviewer that in real data most errors occur in homopolymer regions. We just argue that insertion and deletion errors within a read should occur at about the same frequency.

Line 206. What reads were used for polishing? The HiFi reads, or Illumina reads? If short reads, they are no longer necessary for polishing HiFi based contigs.

R. The polishing step built in the algorithm uses the same long reads that are used for the assembly process. We agree that polishing with Illumina reads is no longer needed for HiFi reads.

Also in this section of the paper, should mention that these assemblies are only contigs, not scaffolds.

R. We clarified that the result of the consensus are contigs

Need to mention that CHM13 is not truly a diploid genome; it is effectively haploid, making the assembly process easier. Also need to say that the reference being compared is the T2T complete CHM13 genome.

R. We clarified in the results that the CHM13 cell line is haploid. We also clarified in the methods that the reference genome for comparison is the T2T assembly. We made sure that we were using the latest version of the T2T assembly as the gold-standard for this comparison.

Line 423: Should cite Cheng et al 2022 Nature Biotech also for HiFiasm Hi-C, and Kronenberg et al 2021 Nature Communications for the first algorithm that used Hi-C for haplotype phasing.

R. We included these papers in the references and cited them in the introduction and in the discussion.

Line 446: Cite Jarvis et al 2022 bioRxiv for the source of the HG002 data. It might be good to also used the genome in this study as the highest quality HG002 reference at this point in time.

R. Based on the comments of the two reviewers, we improved the benchmark of HG002 using this recently published assembly. We cited the paper (recently published in Nature) and improved the results and the discussion with the main outcomes of this comparison.

Reviewer #2 (Comments to the Authors (Required)):

The manuscript describes new algorithms for long-read assembly, which are implemented as a part of the existing NGSEP package. De novo assembly is indeed a key algorithmic challenge in computational biology and any improvements in this field are highly appreciated. The authors benchmark their approach against the state-of-the-art methods using a variety of PacBio and Oxford Nanopore datasets. On most datasets, NGSEP is an average performer in the most important contiguity metrics. It is nevertheless an interesting orthogonal approach with a few algorithmic novelties and will certainly have its use cases. I am enthusiastic about the manuscript in general, but have a few important requests to ensure that the algorithmic novelties and key comparisons are presented clearly.

R. We thank the reviewer for the assessment of our work. We are glad to see that the algorithmic improvements included in this work are well received. We improved the benchmark experiments and made significant changes to the results and discussion to properly address the comments of the reviewer. Please find below our specific answer to each comment.

1. In the description of the algorithms, the authors provide a fair amount of technical detail. It makes it somewhat difficult to understand what are key algorithmic novelties, compared to the existing assemblers. I encourage the authors to provide shorter summaries for non-expert readers that highlight the key differences compared to the other approaches, and the best use case scenarios.

R. We summarized the main algorithmic improvements in the first paragraph of the discussion to highlight better the novelties implemented in our software

2. The N50 metric depends on the assembly size and may be inflated for assemblies with lower genome coverage. It should be substituted with the NG50 metric which does not depend on the assembly size.

R. We used N50 because there were no major differences in the assembly length between tools and because we have full Nx values (N10 to N90) for this statistic. There were a few specific cases where the NG50 was higher than the N50, and all of them were the result of assemblies with total length larger than the reference genomes. This phenomenon was observed in Canu for all datasets, and Hifiasm for the rice datasets. We plotted NG50 values in the supplementary figures 1 and 4 and discussed the values in the results.

3. In addition, I encourage the authors to provide the NGA50 metric, which accounts for possible structural inaccuracies. The authors report NGA50 values in the supplementary tables, but they should be mentioned in the main text (and perhaps added to the NG50 plots).

R. To improve the comparison of the values included in the supplementary tables, we included supplementary figures showing the N50, NG50, NA50 and NGA50 values (See supplementary figures 1 and 4). We also improved the results describing the main results of the analysis of NGA50 values. For the main figures we used Nx values because there were no major differences in the assembly length between tools and because the number of misassemblies is reported separately.

4. According to the methods section, the authors ran QUAST with default parameters, which are optimized for bacterial genomes. The large genome mode "--large" should be used instead for all datasets. This mode accounts for possible structural differences between reference and assembled genomes (as mentioned by the authors) and may substantially affect misassemblies counts, NGA50 and other alignment-based statistics.

R. We thank the reviewer for this suggestion. We looked in more depth the results of Quast and performed a few experiments changing parameters. We noticed that the default parameters generated a severely inflated number of misassemblies for most comparisons. In particular we found that small alternative alignments with relatively low (about 95%) identity were used to mark "misassemblies" in large contigs. Hence, we ran Quast again with the following parameters, mostly consistent with the `--large` option: `--eukaryote --min-contig 25000 --min-identity 99 (98 for ONT data) --min-alignment 5000 --extensive-mis-size 20000`. We updated the figures and tables to report the new statistics and we adjusted the text of the results and discussion according to the new results.

5. The main text (and perhaps figures) should also contain information about the base-level assembly quality (number of mismatches and small indels).

R. We improved the results and we added supplementary figures to compare the base-level assembly quality of the different tools (See supplementary figure 2 and 5).

6. The authors rely on the misassemblies count to argue that their approach is more accurate, however this metric could be misleading. Misassemblies are often clustered together inside difficult-to-assemble regions (e.g. segmental duplications), and reflect the differences between reference and assembled genomes or alignment difficulties. As a result, assemblies that do not

attempt to reconstruct such regions will have less reported misassemblies without significant penalties in completeness metrics. One important change to alleviate this is to use the correct QUAST parameters (as mentioned above). A good strategy is to mask the difficult-to-profile regions of the genome (see the Shasta paper for example), but this may be difficult to do for non-human genomes. An alternative strategy may be to focus only on very large misassemblies (10, 20, 50kb+). The minimum misassembly size should be reported in the main text.

R. Based on the analysis of the Quast results, we found that the parameters that we were using generated an inflated number of misassembly errors. Unfortunately, the NGA50 values are artificially reduced as a consequence of this issue. We adjusted the parameters following the recommendation of the reviewer and in particular we set the limit to call a misassembly as global to 20 kbp, to facilitate focusing on large misassemblies (bold colors in Figure 3 and Figure 5). Although the numbers of mis-assemblies are better now, we still believe that these are higher than those suggested by our internal evaluation of direct minimap alignments. We finally decided to report the results as obtained with the parameters fixed at this time, to ensure that all tools are evaluated using a fair procedure.

7. For the diploid assembly evaluations, the authors should report hamming and switch errors, since the usual contiguity metrics are not sensitive to phasing errors.

R. We improved the benchmark of HG002 using the recently published assembly of HG002 (Jarvis et al., 2022) and expanding the tools to evaluate and the tools used to perform the evaluation (See supplementary table 2 for details). We reported the number of mismatch and indel errors calculated by Quast. This tool was executed with the parameters discussed above for large genomes to have a better assessment of misassembly errors. Understanding that some of the errors reported by Quast can be switch errors between homologous haplotypes, we ran the software Merqury to estimate the total number of switch errors. Although the results were sound comparing tools, in absolute numbers, the amount of switch errors reported by Merqury was over 10 times the amount of misassemblies reported by Quast. Hence, we decided to implement an alternative window-based analysis in which windows are assigned to parental haplotypes based on a selection of parent-specific k-mers from the k-mer distribution obtained with NGSEP. The results of our procedure were consistent with those of Merqury in terms of ranking but the absolute numbers were more consistent with the misassemblies reported by Quast. Finally, we also improved the analysis of genes mapped to the generated assemblies implementing a script to analyze the paf alignment files generated by minimap2. We obtained results that were more consistent with the genome coverages reported by Quast. We completely rewrote the results section on HG002 to describe the new outcomes of our experiments.

8. NGA50 for the phased HG002 was much lower (1.68 Mb), in comparison to the primary hifiasm mode (69 Mb). However I would expect the phased hifiasm mode to have much lower contiguity than currently reported. Did the authors run hifiasm using only HiFi reads, or was this assembly generated with the help of HiC/trio? The comparison with HiFi only phased assembly would be more relevant.

R. We repeated this assembly to double check that the comparison was performed against the primary assembly and the phased assembly of Hifiasm obtained taking as input only HiFi reads. We did not include Hi-C or trio data, although, as the first reviewer indicates and as reported by Jarvis et al. 2022, the inclusion of these data would for sure improve the genome assembly achieved by Hifiasm. We were also surprised by the high N50 values and good quality obtained with the current version of Hifiasm. However, we must be honest in reporting the current behavior of each tool regardless of the results obtained with NGSEP. Despite the current difference in contiguity, we believe that the phasing algorithm implemented in NGSEP is very promising and we are motivated to perform further improvements to improve the N50 and genome coverage. Based on the results of the simulations and on some internal experiments with Trypanosoma samples, we also believe that it is important to develop further benchmark datasets, hopefully for different species, to see how the results obtained with Hifiasm hold on genomes with different compositions of repetitive elements.

9. The authors tested their approach on nanopore data using three relatively small datasets. I suggest providing results for human genomes (CHM13, HG002), for which a lot of data is publicly available. Modern nanopore reads are much longer than PacBio HiFi and should result in longer phased blocks, highlighting a good use case for the authors' method.

R. We thank the reviewer for this suggestion. We ran NGSEP using ultralong Nanopore data from CHM13. After making a couple of improvements in memory management, these ultralong reads provided us with a very contiguous and accurate assembly (N50=71.5 Mbp NGA50=66.04 Mbp). We improved the results to include the experiments with the CHM13 data and to report the metrics as reported for the HiFi experiments. For HG002, the issue to achieve good phasing is that the error correction step to reduce the high error rate effectively eliminates the signal of many heterozygous variants. This makes a big negative impact in the phasing step.

10. The authors need to provide a link to a publicly available repository with the source code and instructions how to run the assembler and reproduce the paper analysis. All assemblies generated in this study should be also made publicly available.

R. We apologize for not including this information in the original manuscript. We included a data and software availability statement. We also added the genome assemblies to the NGSEP sourceforge site (<https://sourceforge.net/projects/ngsep/files/benchmarkAssembler/>).

February 9, 2023

RE: Life Science Alliance Manuscript #LSA-2022-01719-TR

Prof. Jorge Duitama
Universidad de Los Andes
Cra 1 Este # 19 A - 40
Bogotá 111711
Colombia

Dear Dr. Duitama,

Thank you for submitting your revised manuscript entitled "New algorithms for accurate and efficient de-novo genome assembly from long DNA sequencing reads". We would be happy to publish your paper in Life Science Alliance pending final revisions necessary to meet our formatting guidelines.

- please upload your supplementary figure files as single files and make sure that any table files are either included in the doc file of the main manuscript text or upload them as separate editable doc or excel files
- please make sure that all the author names in your manuscript match the author names entered in our system
- please add the author contributions to the main manuscript text
- please add a separate figure legend, including your main and supplementary figures and tables, to your main manuscript text
- please add a figure callout for Figure 5A, Figure 6 and Figure S2 to your main manuscript text

A. FINAL FILES:

B. MANUSCRIPT ORGANIZATION AND FORMATTING:

**Submission of a paper that does not conform to Life Science Alliance guidelines will delay the acceptance of your

manuscript.**

The license to publish form must be signed before your manuscript can be sent to production. A link to the electronic license to publish form will be sent to the corresponding author only. Please take a moment to check your funder requirements.

Sincerely,

Reviewer #2 (Comments to the Authors (Required)):

I thank the authors for responding to all my comments, I have no further requests.

February 13, 2023

RE: Life Science Alliance Manuscript #LSA-2022-01719-TRR

Prof. Jorge Duitama
Universidad de Los Andes
Cra 1 Este # 19 A - 40
Bogotá 111711
Colombia

Dear Dr. Duitama,

Thank you for submitting your Methods entitled "New algorithms for accurate and efficient de-novo genome assembly from long DNA sequencing reads". It is a pleasure to let you know that your manuscript is now accepted for publication in Life Science Alliance. Congratulations on this interesting work.

DISTRIBUTION OF MATERIALS:

Again, congratulations on a very nice paper. I hope you found the review process to be constructive and are pleased with how the manuscript was handled editorially. We look forward to future exciting submissions from your lab.

Sincerely,
